# Short-term effectiveness of nutrition therapy to treat type 2 diabetes in low-income and middle-income countries: systematic review and meta-analysis of randomised controlled trials

Elizabeth Guilbert,[1] Rachel Perry ,[2] Alex Whitmarsh,[2] Sarah Sauchelli [2]

¹Faculty of Health Sciences, University of Bristol, Bristol, UK
²National Institute for Health Research Bristol Biomedical Research Centre; University Hospitals of Bristol and Weston NHS Foundation Trust and University of Bristol, Bristol, UK

**Correspondence to**
Dr Sarah Sauchelli;
sarah.sauchellitoran@bristol.ac.uk

## ABSTRACT

**Objectives** This review examined the evidence arising from randomised controlled trials regarding the impact of nutrition therapy on glycaemic control in people living with type 2 diabetes mellitus (T2DM) in low/middle-income countries (LMICs).

**Design** Systematic review and meta-analysis using the Grading of Recommendations, Assessment, Development and Evaluation (GRADE) Approach.

**Data sources** MEDLINE, EMBASE, Web of Science, OpenGrey and the International Clinical Trials Registry were searched (up to July 3 2020).

**Eligibility criteria** Trials were included if they evaluated nutrition therapy in adults diagnosed with T2DM, were conducted in LMICs, measured glycaemic control and the trial included a 3-month post-intervention assessment. Nutrition therapy was defined according to American Diabetes Association recommendations.

**Data extraction and synthesis** Two independent reviewers screened the database. Study characteristics and outcome data were extracted using a data collection form. Meta-analyses were conducted for glycated haemoglobin (HbA1c) and fasting blood glucose. Trials were assessed for risk of bias (Cochrane Risk-of-Bias, Version 2.0) and overall certainty of evidence (GRADE).

**Results** Four trials met inclusion criteria (total n=463), conducted in Malaysia, Iran and South Africa. All trials focused on nutrition education with no direct prescription or manipulation of diet. Mean differences between intervention and standard care were −0.63 (95% CI −1.47% to 0.21%) for HbA1c and −13.63 mg/dL (95% CI −37.61 to 10.34) for fasting blood glucose in favour of the intervention. Given the small number of eligible trials, moderate to high risk of publication bias and serious concerns regarding consistency and precision of the evidence, certainty of evidence was deemed to be very low.

**Conclusions** There is a lack of well-conducted randomised controlled trials that examine the long-term impact of nutrition therapy in LMICs, preventing firm conclusions to be made on their effectiveness. Further research is essential to discover realistic, evidence-based solutions.

**PROSPERO registration number** CRD42020188435.

### Strengths and limitations of this study

► This is the first synthesis of evidence for the effectiveness of nutrition therapy for the management of type 2 diabetes mellitus in low/middle-income-countries derived from randomised controlled trials that have included a minimum of a 3 month post-intervention assessment.

► The review was conducted following BMJ Best Practice research methods, such as the use of Grading of Recommendations, Assessment, Development and Evaluations to provide a transparent critical appraisal of the certainty of evidence.

► The identification of few trials that met eligibility criteria highlights the need to build capacity for high-quality and long-term research to guide context appropriate evidence-based medicine, and to consider pragmatic trials as alternative research methodologies to gather evidence.

► The focus on randomised controlled trials that included post-intervention assessment signifies that research on other interventions that adopted a different study design was not captured.

## INTRODUCTION

Type 2 diabetes mellitus (T2DM) is rapidly becoming a global pandemic, currently affecting 8.8% of the world's population.[1] It causes significant suffering for those affected, increases mortality and costs the global economy US$727 billion per year.[1 2] Additionally, diabetes is no longer a disease of wealthy nations. Approximately 79% of the 254 million people affected live in low/middle-income countries (LMICs).[1 3–5] The rising prevalence of T2DM in LMICs is particularly problematic since these countries have limited resources to implement effective interventions that are sustainable over time[4 6] and to treat the chronic comorbidities linked to T2DM.[7]

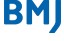

LMICs display the fastest increase in the contribution of the global burden of diabetes.[2] Epidemiological studies show that development and urbanisation in LMICs have led to a less active lifestyle[8 9] and an increased consumption of processed foods and sugary snacks with a high glycaemic index.[3 10] Combined with an improvement in infectious disease management and population ageing,[11] societal changes have expanded the proportional contribution of diabetes to the global burden of disease.[3 12] Nutritional interventions can offer a highly cost-effective avenue to address T2DM.[6]

Nutrition therapy, which promotes healthy eating patterns by targeting nutrient intake and/or portion size,[13] can ameliorate glycaemic control, weight management and delay complications of diabetes.[14–17] Good glycaemic regulation can also aid prevention of cardiovascular disease, which is the greatest cause of morbidity and mortality in T2DM.[6 16] These effects have also been observed in the absence of pharmacological agents, when people are unable to travel to healthcare facilities or when there is a lack of education and health literacy.[6 10 14] However, high-quality research on T2DM prevention and management strategies has been primarily carried out in high-income countries (HICs).[5] Randomised controlled trials (RCTs), considered as a gold standard for evidence-based health interventions,[18] are less common in LMICs, with 78% of RCTs for non-communicable diseases recruiting participants in HICs.[19]

Given the scarce number of trials available in many LMICs, development and implementation of interventions lacks rigorous scientific appraisal, or will often be based on the evidence gathered in HICs. Direct application of interventions tested in HICs may not be effective in LMICs, as they may not be sensitive to context-specific characteristics of communities in LMICs in terms of religion, socioeconomic status, composition of staple foods and cultural norms.[7] These factors strongly influence both prevalence and management of T2DM and hence the success of nutritional interventions.[1 14 20] In addition, assessment of outcomes following the termination of an intervention is fundamental to evaluate its effectiveness. Weight regain is frequent in people with T2DM, particularly when continuous support provided in weight loss interventions ends.[21] Measurement of glycated haemoglobin (HbA1c), the standard measure of glucose regulation, reflects the average blood glucose levels over the 2–3 months prior to measurement. Therefore, when evaluating the effects of an intervention on glycaemic control, it is essential that a 3-month follow-up assessment is in place. This is likely to be rare in LMICs, when critical analysis of evidence from HICs reveals that long-term follow-up is often limited to a maximum of 8 months post-intervention.[17 22 23]

The purpose of this study was to systematically review RCTs carried out in LMICs assessing the short-term effectiveness of interventions, focusing on nutrition therapy to improve glycaemic control in people with T2DM. Nutrition therapy was defined according to the consensus recommendations presented by the American Diabetes Association for diabetes and pre-diabetes,[13] where treatment entails the modification of nutrient or whole-food intake. This definition includes interventions that provide individualised, diabetes-focussed management plans and ongoing monitoring, those that address individual nutrition needs and that distribute positive messaging about food choices, as well as nutrition education, lifestyle intervention programmes with goal-setting, and provision of tools for day-to-day meal planning. Importantly, the review will consider those interventions that have included a 3-month post-intervention assessment of HbA1c, to assess the primary outcome measure of potential benefits on glycaemic control following intervention completion. We expected that the present review would add to an earlier analysis of nutrition interventions for T2DM prevention in LMICs,[24] thus providing an overview of high-quality research conducted to address the management of T2DM in LMICs.

## METHODS

### Search strategy and selection criteria

Following the steps laid out in our protocol (PROSPERO CRD42020188435), a systematic literature search was conducted to identify RCTs carried out in LMICs as defined by the Development Assistance Committee list of recipients of Official Development Assistance published in 2018.[25] No amendments were made. Throughout this process the Preferred Reporting Items for Systematic Reviews and Meta-Analyses (PRISMA) Checklist was used to ensure proper reporting of the review.[26] The databases MEDLINE, EMBASE and Web of Science were searched, as well as OpenGrey and the International Clinical Trials Registry Platform (ICTRP). The search was completed on July 3 2020. Search terms can broadly be divided into four categories: diabetes, nutrition, diet, LMICs and RCTs (see online supplemental file 1 for the Medline search). The references of relevant systematic reviews identified through literature searching were manually checked for further papers, as were the reference lists of included papers. Authors of unpublished clinical trials that were included based on protocols, were contacted by email to request results or manuscripts where available.

The inclusion and exclusion criteria followed the PICO framework.[27]

### Types of studies

▶ Published and unpublished RCTs that investigate nutrition therapy for T2DM.
▶ Minimum follow-up period of 12 weeks from the end of intervention.
▶ Full-text reports in any language.

### Participants

▶ Adults (≥18 years), male and female, any ethnicity.
▶ Residents of a low-income, middle-income or upper middle-income country as defined by the

Development Assistance Committee list of recipients of Official Development Assistance for 2018–2020.

► Diagnosis of T2DM based on HbA1c ≥6.5%, random blood glucose test or oral glucose tolerance test ≥11.1 mmol/L.

► Exclusion: pre-diabetes, metabolic syndrome without definitive T2DM diagnosis, type 1 diabetes or gestational diabetes.

### Interventions

► Nutrition interventions delivered in person, in groups, through mobile/internet-based services or any other means.

► Food supplements/replacements.

► Calorie/diet restrictions.

► Lifestyle education interventions with a focus on diet.

► Exclusion: pharmacological interventions, preventative interventions.

### Comparators

► No intervention/standard or minimal care/intervention that does not include nutrition therapy.

### Outcomes

► Primary: glycaemic control (including HbA1c, fasting plasma glucose, insulin sensitivity/resistance).

► Secondary:
  – Symptoms, for example, reduction in polyuria, polydipsia, fatigue.
  – Diabetic complications, for example, cardiovascular events, retinopathy, diabetic foot, nephropathy, neuropathy, hypo-glycaemia and hyper-glycaemia.
  – Psychological effects including quality of life and enjoyment of food.
  – Adverse effects including malnutrition and economic consequences of diet.

► Quantitative outcomes only, either continuous or categorical

Screening was not limited by date or language and both published and unpublished work was reviewed. In LMICs, interventions must be able to be maintained once funding and services provided by the investigators have been withdrawn. In addition, HbA1c levels reflect average blood glucose levels over the 2–3 months prior to measurement. For this reason, trials were excluded if they had less than a 12-week follow-up period after the intervention had finished. Those trials where assessment was only carried out while the intervention was ongoing were excluded. Trials of a population with gestational or type 1 diabetes were also excluded as management strategies differ from those for people with T2DM.[13 28] Trials were included when the main component was nutrition therapy (ie, diet modification) achieved by directly prescribing a diet, meal replacement or use of supplements or by encouraging change via nutrition education. Where nutrition therapy was not the most influential component expected to contribute to changes in glycaemic control (eg, one structured nutrition education session within a 2-year exercise programme, or a short course of meal replacement combined with long-term metformin prescription), the RCT was excluded.

The trials identified were imported into Rayyan[29] for abstract screening by two independent researchers (LG and SS) after duplicates had been removed using EndNote Web. The same two researchers (LG and SS) independently screened full-texts. Conflicts between researchers were resolved by consensus, with the support of a third researcher when necessary (RP).

### Data analysis

Based on the inclusion and exclusion criteria, four papers were included after full-text screening. Data were collected by LG and checked by SS using an adapted version of the Cochrane data collection form for RCTs and non-randomised studies.[30] Data on HbA1c and fasting plasma glucose were extracted. Where HbA1c and/or fasting plasma glucose levels were reported in different measuring units across papers, mean and SD of results were converted to achieve a comparable set of results (eg, mmol/L to mg/dL; See online supplemental file 2 for conversion). Other domains of extraction included study design characteristics, population, intervention, comparator and secondary health outcomes. Outcomes extracted were the ones reporting glucose control data after completion of the intervention, which could be at 3 months or 6 months. The data extraction form included a template for Risk of Bias assessment based on version 2.0 of the Cochrane Risk-of-Bias instrument for randomised trials. Each domain was assessed to produce scores of 'high' or 'low' risk or 'some concerns'. Data extraction was conducted independently and in duplicate. Where information was missing, attempts were made to obtain it from the authors.

Meta-analyses were conducted using Stata V.16 (StataCorp LLC) for the two primary outcomes (HbA1c and fasting blood glucose). Pooled mean differences were calculated with corresponding 95% Confidence Interval (CI). First, we pooled trials according to the time at which the post-intervention assessment was undertaken (3–6 months), and performed separate analyses at both 3 months and 6 months. Second, we generated an overall effect estimate. As we expected both clinical and methodological variation between eligible studies of which cause would be difficult to identify, random effect analyses were considered as most appropriate. Results were presented in the form of forest plots.

Overall certainty of evidence was rated using the Grading of Recommendations, Assessment, Development and Evaluation (GRADE)[31] approach, where RCTs begin at the highest level of certainty and are downgraded if concerns arise in one or more of the following five domains: risk of bias, inconsistency, indirectness, imprecision and publication bias.[32 33] GRADE was completed independently by SS and RP, and consensus was reached with oversight from AW. Dispersion of true effect sizes was reported using the $T^2$ statistic, and the proportion of

variance in point estimates due to variation was measured using the $I^2$ statistic. The $\chi^2$ test was used for homogeneity. Adhering to recommendations in the Cochrane Handbook, an $I^2$ statistic of 50%–90% was considered as substantial impact, and a score of 75%–100% was deemed evidence for considerable inconsistency.[34] As the number of eligible trials was inferior to 10, we did not proceed with using a funnel plot to evaluate publication bias, as originally planned. Trials were scrutinised individually following recommendations by Guyatt and colleagues[31] to provide a final judgement of 'strongly suspected' or 'undetected' on the GRADEpro software tool.

### Data interpretation

There are no clear-cut thresholds to conclude that an intervention under evaluation is superior to standard care on the basis of HbA1c or fasting blood glucose. Previous work suggests that a reduction by 0.5% or 1% in HbA1c is often used by health professionals when making adjustments to therapy[35] and is beneficial for reducing cardiovascular disease risk, a patient important outcome.[36] Given that fasting blood glucose is often a secondary end point in RCTs, establishing a threshold for meaningful effect was also derived from vascular risk, which suggests a threshold value of 18 mg/L.[37]

### Patient and public involvement

No patients were involved in this research study. Informal discussions with health professionals in Cameroon as part of another project raised the need to conduct the systematic review before co-designing a novel intervention with their patients.

### RESULTS

The initial literature search yielded 5075 results. After duplicates were removed, 1948 records remained. Title and abstract screening of these records was carried out by two reviewers and gave a percentage agreement of 91% and a Cohen's kappa statistic of k=0.45 indicating moderate agreement.[38] Eighteen trial protocols were identified as potentially relevant for which authors were contacted for results. Of these, 14 did not provide a response, two had not yet completed analysis, and two returned a published manuscript. After conflicting decisions were resolved, 44 trials remained for full-text eligibility checking, and four trials met inclusion criteria (see figure 1). One of the most common reasons for exclusion was the absence of a 3-month post-intervention assessment.[39]

The four included papers (table 1) reported trials conducted in South Africa,[40] Malaysia[41] and Iran,[42 43] which are all upper-middle income countries (a subgroup of LMICs) according to the Organisation for Economic Co-operation and Development.[25] Combined, the trials examined a sample size of 463 participants (mean study sample: 115.75, weighted SD of sample size: 27.11). Two hundred and thirty-four participants had been assigned to the intervention arm, and 229 to the control arm. All

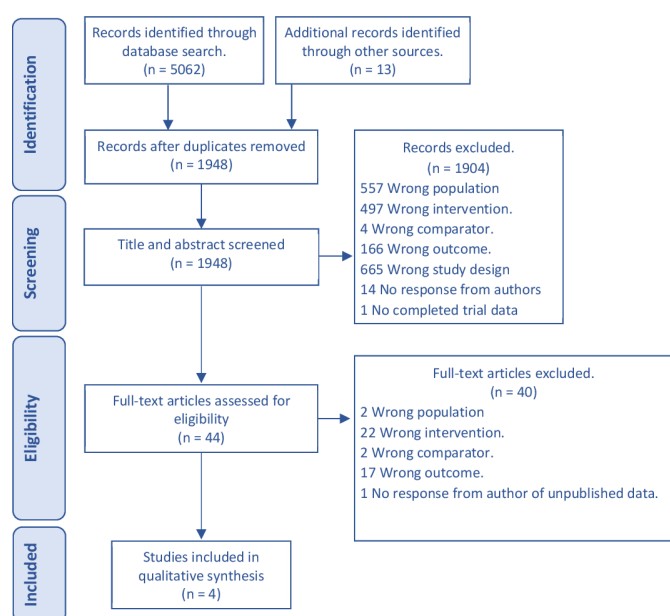

**Figure 1** PRISMA study flow. PRISMA, Preferred Reporting Items for Systematic Reviews and Meta-Analyses.

trials recruited participants from medical or diabetic clinics; two were hospital based and two from community health centres.

All participants had been diagnosed with T2DM, had a weighted mean age of 57 years (SD: 6.4 years), and 63% (n=291) were women. One trial reported the presence of comorbidities (hypertension, dyslipidaemia and a history of cardiovascular disease), one excluded those candidates with chronic conditions in addition to T2DM, and two trials did not provide information on comorbidity. The baseline weighted mean HbA1c of participants was 8.75% (SD: 1.38%) collected from the four trials, and the baseline weighted mean fasting blood glucose was of 154.73 mg/dL (SD: 5.41 mg/dL) collected from three trials.

All RCTs focused on nutrition education to prompt people to make healthy dietary choices and improve portion control. There was no direct manipulation of nutrient or whole-food intake (eg, prescribing a specific diet, meal replacement or providing food supplements/substitutes). In one trial, participants received a pamphlet and fridge/wall poster in line with standard care, complemented by an 8-week face-to-face group (6–8 people) nutrition education programme (dietitian-led) that targeted knowledge deficits and inappropriate dietary practices, and six follow-up sessions.[40] The eight sessions covered the following topics: definition of diabetes, treatment, dietary guidelines (two sessions), gardening to increase vegetable supply, meal planning (two sessions) and meal preparation.[40] The second trial provided dietary lesson plans (12 sessions, fortnightly) delivered via a website and tailored to each participant's current Dietary Stage of Change (a system to classify an individual's readiness to change a health-related behaviour).[41] The sessions included tailored recommendations aimed at improving

**Table 1** Characteristics of included trials

| Trial (year) | Population: age group; diagnosis; geographical setting; care setting | Mean age (years): intervention/control; % female | Intervention; duration | Comparator | Timing of assessment | Outcomes assessed |
|---|---|---|---|---|---|---|
| Askari (2018)[43] | Age >18years; T2DM; Iran (UMIC); diabetes centre. | 66/67; 65% female | Face-to-face training in diabetes, diet and the importance of exercise (8 sessions over 4 weeks) with weekly reminder texts, instructions to walk at least three times per week, two post-intervention calls and pamphlets given to family and relatives. | Standard care | Baseline, 3 months after intervention completion. | HbA1c (%), fasting blood glucose (mg/dL), LDL and HDL lipoprotein-cholesterol (mg/dL), triglycerides (mg/dL). |
| Muchiri (2016)[40] | Age >18years; T2DM; South Africa (UMIC); diabetic clinics in two community health centres. | 59/58; 87% female | Face-to-face group tutorials and six follow-up sessions on diabetes diet, meal planning, portion control and gardening to improve vegetable intake; 8 weeks. | Standard care. Pamphlet and poster | Baseline, at intervention completion, 6 and 12 months after intervention completion. | HbA1c (%), BMI (kg/m$^2$), total, LDL and HDL lipoprotein-cholesterol, triglycerides (mmol/L); systolic blood pressure (mm Hg), diastolic blood pressure (mm Hg), energy intake (kJ/day), % g/day of intake from carbohydrate, protein, fat, alcohol, SFA (% of energy), PUFA (% of energy) and MUFA (% of energy), servings of vegetables (servings/day), fruit (servings/day), starchy foods (servings/day). |
| Ramadas (2018)[41] | Age >18years; T2DM; Malaysia (UMIC); medical or diabetic clinics at three public hospitals. | 50/52; 39.8% female | Personalised online dietary lessons tailored to the patient's status and recommendations to increase fruit and vegetable consumption, staying healthy and reducing sugar intake; 12 sessions. | Standard care | Baseline, at intervention completion, 6 months after intervention completion. | HbA1c (%), fasting blood glucose (mmol/L) DKAB questionnaire (knowledge, attitude and behaviour score). |
| Salahshouri (2018)[42] | Age >18years; T2DM; Iran (UMIC); diabetic clinics at healthcare centre. | 56/55; 69.3% female | Face-to-face group sessions targeting perceptions, beliefs, concerns and discomfort around a diabetic diet; 8 weeks. | Standard care | Baseline, 3 months after intervention completion. | HbA1c (%), fasting blood glucose (assumed mg/dL), 4-part questionnaire comprising: (a) demographic characteristics, (b) nutrition perceptions and beliefs, (c) fears, concerns and discomfort as about nutrition diet, (d) perceived Dietary Adherence Questionnaire. |

BMI, body mass index; DKAB, Diabetes Knowledge, Attitude and Behaviour; HbA1c, glycated haemoglobin; HDL, High-density lipoprotein; LDL, Low-density lipoprotein; MUFA, Monounsaturated fatty acid; PUFA, Polyunsaturated fatty acid; SFA, saturated fatty acid; T2DM, type 2 diabetes mellitus; UMIC, Upper- to Middle-Income Country.

diabetes knowledge and behaviour, address barriers to dietary change and motivate participants. The third trial provided educational sessions delivered by a range of health specialists (eg, dietitians, psychologists) in the meeting hall of a healthcare centre. These were aimed at addressing perceptions, beliefs, fears, concerns and discomforts towards a diabetic diet (eg, based on psychological factors that determine nutrition such as avoiding temptation, communication skills training, replacing irrational thinking, religious laws around health, and focussing on success) in the form of eight 1-hour sessions with groups of 23–24 participants.[42] The fourth trial delivered a group educational intervention over 4 weeks (two 70 min training sessions per week) based on the belief, attitudes, subjective norm and enabling factors model; giving participants in the intervention arm information about diabetes, symptoms of hyperglycaemia and hypoglycaemia, diet, use of food composition tables, partitioning and food replacement and appropriate intake of fruits, vegetables and grains.[43] This trial was the only one to ask participants to record daily food consumption (though the use of those records could not be identified), conduct follow-up calls (two calls), send weekly reminder text messages and to include exercise in the form of a prescribed jogging regime.[43]

The common outcome used to assess glycaemic control was HbA1c, and three trials additionally measured fasting blood glucose. Assessment of glycaemic control took place at baseline across all trials and one trial at end of intervention. Two trials reported post-intervention data on glycaemic control at 3 months, and two at 6 months. One study had an additional post-intervention assessment after 12 months.[40] Dropouts were present across all interventions. Overall, both HbA1c and fasting blood glucose were found to have unclear risk of bias in one study and evidence for high risk of bias in one (figure 2). Low risk of bias was deemed in the measurement of the outcomes due to the standardised procedure adopted to measure glycaemic control. However, two trials lacked sufficient details on participant allocation and the absence of a priori protocol prevented the assessment of selective reporting bias. In one trial,[43] participants were excluded if they missed two sessions of training, but there was no indication on the number of participants who were therefore also withdrawn from the trial. A 'high risk' judgement was made in regards to complete accounting of patients.

Table 2 provides a summary of findings for primary outcomes. Figure 3 presents the pooled analysis for HbA1c and fasting blood glucose. As two trials measured these outcomes at 3 months after intervention completion and two at 6 months, pooled analyses were split by timepoint of assessment. At 3 months, the nutrition interventions were more effective than standard care (control), with the pooled mean differences at –1.11% (95% CI –1.64% to –0.59%) for HbA1c and –23.57 mg/dL (95% CI –44.3 to –2.84) for fasting blood glucose. The advantage was less clear in trials with a 6 month post-intervention assessment, with pooled mean differences at –0.09% (95% CI –1.10% to 0.91%) for HbA1c and 9 mg/dL (95% CI –7.55 to 25.55) for fasting blood glucose. Similar uncertainty is present when examining all trials jointly, with pooled mean differences at –0.63% (95% CI –1.47% to 0.21%; four trials, n=463, GRADE=Very low) for HbA1c and –13.63 mg/dL (95% CI –37.61 to 10.34; three trials, n=381, GRADE=Very low) for fasting blood glucose. The point estimates for both HbA1c and fasting blood glucose include both the line of 'no effect' and the threshold for concluding clinically meaningful difference.

Note, HbA1c provides an average measure of blood glucose over the previous 3 months, due to the lifecycle of red blood cells. Further, the effects of nutrition interventions are likely to be sustained in the early months post-intervention. We would therefore not expect the difference in timepoint assessment to have a clinically meaningful impact on the reported effects. The small number of trials that met inclusion criteria does not permit definite conclusions to be reached.

Secondary outcomes are presented in table 3. Briefly, one trial reported[42] a notable improvement in perceived adherence to a healthy diet 3 months post-intervention, whereas another trial reported the intervention to generate benefits for dietary knowledge, attitudes and behaviour at 6 months.[41] Two trials further examined the impact of nutrition education on lipid profile and on body mass index (BMI), biomarkers of elevated risk of diabetes.[44] High-density lipoprotein cholesterol appears to improve following the intervention when the post-intervention assessment occurs at 3 months,[43] but this is no longer so when it occurs at 6 months after intervention end.[40] After 3–6 months after intervention completion, nutrition education does not seem to impact other markers of increased risk of diabetes complications (eg,

| Outcome: HbA1c | Randomization process | Deviations from intended interventions | Missing outcome data | Measurement of the outcome | Selection of the reported result | Overall bias |
|---|---|---|---|---|---|---|
| Askari (2018)[43] | Some concerns | Some concerns | High risk | Low risk | Some concerns | High Risk |
| Muchiri (2016)[40] | Low risk | Low risk | Low risk | Low risk | Low risk | Low Risk |
| Ramadas (2018)[41] | Low risk | Low risk | Low risk | Low risk | Low risk | Low Risk |
| Salahshouri (2018)[42] | Some concerns | Low risk | Some concerns | Low risk | Some concerns | Some concerns |

| Outcome: Fasting blood glucose | Randomization | Deviations from intended interventions | Missing outcome data | Measurement of the outcome | Selection of the reported result | Overall bias |
|---|---|---|---|---|---|---|
| Askari (2018)[43] | Some concerns | Low risk | High risk | Low risk | Some concerns | High Risk |
| Muchiri (2016)[40] | N/A | N/A | N/A | N/A | N/A | N/A |
| Ramadas (2018)[41] | Low risk | Low risk | Low risk | Low risk | Low risk | Low Risk |
| Salahshouri (2018)[42] | Some concerns | Low risk | Some concerns | Low risk | Some concerns | Some concerns |

**Figure 2** Risk of bias of individual trials, by primary outcome measure. HbA1c, glycated haemoglobin.

**Table 2** Summary of findings for primary outcomes

| Outcomes | Anticipated absolute effects* (95% CI) | | Relative effect (95% CI) | No of participants (trials) | Certainty of the evidence (Grade) |
| --- | --- | --- | --- | --- | --- |
| | Risk with standard care | Risk with nutrition therapy | | | |
| HbA1c Scale from: 0% to 100% follow-up: range 3 months to 6 months | The mean HbA1c ranged from 8.13% to 10.3% | MD 0.63% lower (1.47 lower to 0.21 higher) | – | 463 (4 RCTs) | ⊕◯◯◯ Very low†‡§¶ |
| Fasting blood glucose follow-up: range 3 months to 6 months | The mean fasting blood glucose ranged from 136.8 to 153.64 mg/dL | MD 13.63 mg/dL lower (37.61 lower to 10.34 higher) | – | 381 (3 RCTs) | ⊕◯◯◯Very low†¶**†† |

GRADE Working Group grades of evidence: High certainty: we are very confident that the true effect lies close to that of the estimate of the effect. Moderate certainty: we are moderately confident in the effect estimate: The true effect is likely to be close to the estimate of the effect, but there is a possibility that it is substantially different. Low certainty: our confidence in the effect estimate is limited: the true effect may be substantially different from the estimate of the effect. Very low certainty: we have very little confidence in the effect estimate: the true effect is likely to be substantially different from the estimate of effect.

*The risk in the intervention group (and its 95% CI) is based on the assumed risk in the comparison group and the relative effect of the intervention (and its 95% CI).

†Rated down for the small number of trials. Two trials additionally presented unclear or high risk in several domains: randomisation process, missing outcome data and selection of the reported result.

‡Heterogeneity: p=0.01. Downgrading carried out because p<0.01 is considered as the threshold for consistency.[59] Small sample sizes, with an I² of 76%.

§The optimal information size (OIS) is 106 (based on a power of 0.80 being considered as adequate), and the total sample size of the analysis is above this (n=463). However, rated down because the CIs of two trials contain the null effect, and though there is no harm from the intervention, clinical judgement suggests that an absolute effect size of 0.63 is not sufficiently high to judge in favour of the nutrition intervention.

¶Rated down because of the small sample size of the trials, the inability to access some full text manuscripts that might have met the inclusion criteria, and likely existence of ongoing clinical trials that may have been missed.

**Heterogeneity: p=0.000. Downgrading carried out because p<0.01 is considered as the threshold for consistency.[59] Small sample sizes, with an I² of 92%.

††The optimal information size (OIS) is 730, the total sample size of included trials is below this value (n=381). Further the CIs do not exclude no effect (−7.55 to 25.55).

HbA1c, glycated haemoglobin; MD, mean difference; RCTs, randomised controlled trials.

a)

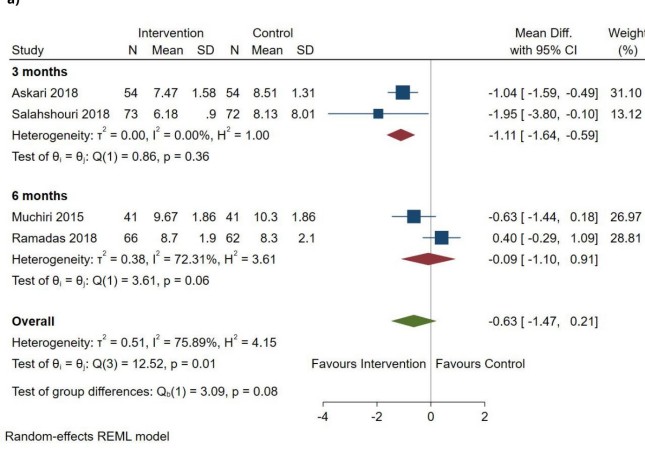

Random-effects REML model

b)

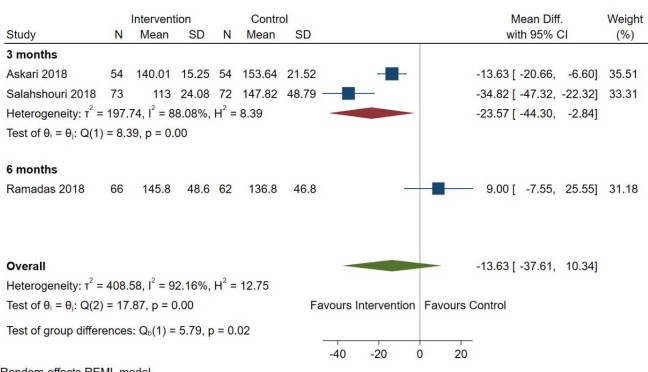

Random-effects REML model

**Figure 3** Change in (A) HbA1c at follow-up assessment, split according to timepoint of assessment (3 or 6 months) and overall; and (B) fasting blood glucose at follow-up assessment, split according to timepoint of assessment (3 or 6 months) and overall. HbA1c, glycated haemoglobin.

heart disease) evaluated by the trials. However, the quality of the evidence is overall very poor (Grade=Very low). No further information was reported on adverse events nor other pre-specified secondary outcomes.

Apart from grouping analyses according to timepoint of post-intervention assessment, no further subgroup analyses were conducted given the limited number of trials eligible for inclusion. Tables 1–3 present GRADE judgements for primary outcomes and secondary outcomes respectively. The certainty of evidence on the short-term benefits of nutrition education for treating T2DM in LMICs is overall very low. This is primarily due to the small number of RCTs with at least a 3-month post-intervention assessment conducted in these countries, the inconsistency between trials, high level of imprecisions and likelihood of publication bias. Quality of evidence regarding other types of nutrition therapy could not be judged due to an absence of RCTs meeting our eligibility criteria.

Online supplemental file 3 presents the completed PRISMA checklist.

## DISCUSSION

This systematic review set out to synthesise the evidence on the short-term effectiveness of nutrition therapy for glycaemic control in people living with T2DM in LMICs. Only four trials met the eligibility criteria. The trials focused on nutrition education, delivered via a web platform or face-to-face group training. The small number of trials, considerable heterogeneity and concerns over the methodology and reporting in some trials indicate that evidence available from RCTs at present is of very low quality. Given the very low certainty of the evidence and point estimates, the added benefit of nutrition education

**Table 3** Table of findings for secondary outcomes

| Outcome | Endpoint | Participants (trials) | Effect estimate (95% CI) | Grade |
|---|---|---|---|---|
| Psychological effect: perceived adherence to a healthy diet | 3 months | 145 (1 trial) | MD in score 17.56 (14.66 to 20.46) | Very low |
| Psychological effect: DKAB total score | 6 months | 128 (1 trial) | MD in total score 5.18 (2.05 to 8.31) | Very low |
| LDL cholesterol | 3 months | 108 (1 trial) | MD −4.62 mg/dL (−9.55 to 0.31) | Very low |
| | 6 months | 82 (1 trial) | MD −3.12 mg/dL (−11.77 to 5.53) | Very low |
| HDL cholesterol | 3 months | 108 (1 trial) | MD 5.79 mg/dL (2.42 to 9.16) | Very low |
| | 6 months | 82 (1 trial) | MD −0.39 mg/dL (−3.15 to 2.37) | Very low |
| Triglycerides | 3 months | 108 (1 trial) | MD −9.70 mg/dL (−22.10 to 2.70) | Very low |
| | 6 months | 82 (1 trial) | MD −17.80 mg/dL (−49.27 to 13.67) | Very low |
| BMI | 6 months | 82 (1 trial) | MD −0.30 kg/m² (−0.85 to 0.25) | Very low |
| Systolic blood pressure | 6 months | 82 (1 trial) | MD 4 mm Hg (−4.04 to 12.04) | Very low |
| Diastolic blood pressure | 6 months | 82 (1 trial) | MD 0.10 mm Hg (−4.06 to 4.26) | Very low |

BMI, body mass index; DKAB, Dietary Knowledge, Attitudes and Behaviour; HDL, High-density lipoprotein; LDL, low-density lipoprotein; MD, mean difference.

on HbA1c levels or fasting blood glucose in people with T2DM living in LMICs is unclear.

Findings from this review seem to contrast positive results on the effects of nutrition therapy obtained from community-based prevention strategies and lifestyle interventions, which have shown to reduce the risk of diabetes[24 45] and improve glycaemic control.[5 6 45] The discrepancy, however, may be attributed to numerous factors. First, there is a lot of heterogeneity in the types of nutritional interventions tested in LMICs,[5] possibly because they have been adapted for the target setting and resources available. As this review examined outcomes at 3 and 6 months after intervention completion (vs at completion) and all eligible trials focused on nutrition education (vs other types of nutrition therapy), comparability of findings is diminished. The emphasis on a 3-month post-intervention assessment is a strength of this review over others, as HbA1c provides an aggregate measure of the previous 3 months. Second, participants' baseline HbA1c levels in three trials were above the 10% threshold used to recommend insulin therapy initiation,[46] or combining lifestyle interventions with pharmacotherapy,[47] but there was no information regarding participants' use of pharmacotherapy. Third, the HbA1c levels of the control group in one study improved quite notably, which could lessen the relative effectiveness of the intervention and have influenced the effect estimate. Nonetheless, the wide CIs in the effect estimate indicate that the true effect may be clinically meaningful and accrual of research could provide increased confidence in estimates.

A key consideration for clinicians and policy makers is that the American Diabetes Association's definition of nutrition therapy encompasses nutrition education and interventions that yield modification of nutrient or whole-food intake.[13] Tailoring provision of nutrition guidance to the individual, delivery by an expert in diabetes care and ongoing monitoring to permit modification as needed have been identified as fundamental features of effective nutrition therapy in type 2 diabetes care.[13] These components of nutrition therapy were not consistently present in identified trials but could strengthen observed effectiveness of nutrition therapy. In line with guidelines generated from research in HIC,[48] diabetes management may also benefit from a multi-faceted approach, with equal importance given to nutrition, physical activity and diabetes education; with medication administered when necessary. Within the context of the low-resourced healthcare systems found in LMICs, it may be tempting to seek cost-saving solutions. Preliminary evidence gathered through this review suggests that more comprehensive nutrition therapy and lifestyle interventions may be necessary to address the growing burden of diabetes in LMICs, particularly given that the positive impact of nutrition education on glycaemic results appears to dissipate 6 months after completion of the intervention.

A weakness of this review is that the small number of trials identified did not permit subgroup analysis by

separating interventions that consisted only in nutrition education from the ones with additional components (eg, follow-up sessions, exercise). Further, risk of bias for each outcome was assessed by considering the intervention delivered in each trial in its entirety. Although we would not expect differences in individual study risk of bias assessments based on the additional components described, future research should consider the possible impact of evaluating a multicomponent intervention (eg, adding nutrition education to pharmacotherapy) on risk of bias.

The small number of trials identified in this review reflects consensus that there is a general absence of large experimental research around diabetes management interventions in LMICs.[5 49] While conventional RCTs are considered the gold standard for evidence-based medicine,[18] which are crucial to ensure maximisation of available healthcare resources,[50] the use of pragmatic trials may be more appropriate to derive evidence in LMICs.[51] Short-term interventions funded by overseas agencies rarely involve local stakeholders and the research findings are less likely to be made available locally, although they may gain more international attention when undertaken by an agency from an HIC.[52] As demonstrated by this review, adopting an RCT design may also not add value to available evidence if it is not of high standard or lacks longer term assessment. It is in countries where clinical outcome assessment after intervention end are least reported where continued assessment is most critical; sustainability is essential if it is to benefit those living in the community under investigation, and if research agendas are to be set according to local needs.[53]

The trials identified in this review benefit from being carried out in three separate countries (Malaysia, South Africa and Iran) which vary greatly in terms of culture and diet. However, all three countries where the trials were run are classified in the upper-middle income country subgroup of LMICs according to Organisation for Economic Co-operation and Development,[25] which reflects other studies showing these countries to produce the most literature related to non-communicable diseases.[19] This may be because these countries have higher prevalence of diabetes as they progress through the epidemiological transition most rapidly, and hence have more relevant literature.[54] It may also be attributable to the inequity in availability and access to medical treatment that is widespread in low-income countries, which remains a problem even as non-communicable disease research increases in LMICs as a whole.[1 19 55] Though funding for nutrition research is increasing for these countries, the majority continues to be directed towards addressing undernutrition.[56]

By using a robust evidence synthesis method, including Cochrane's Risk of Bias instrument version 2.0 and the GRADE approach, this review demonstrates that drawing conclusions from RCTs conducted in these countries may be premature if the quality of the evidence is not carefully examined. Given the short-term nature of the

interventions assessed in the trials that met eligibility criteria, measuring HbA1c levels some time after intervention completion provides a more accurate indication of the impact of these types of interventions on glycaemic control. However, adopting this approach signified that the review did not capture other types of interventions. Another weakness of this review is that we were unable to summarise data on actual diet and physical activity. Although neither were in the pre-registered protocol as we focused on surrogate measures of patient important outcomes (eg, HbA1c, fasting blood glucose and BMI), dietary behaviour and physical activity should be measured and reported by RCTs in this area of research. Tthe information should be measured in RCTs and reported in future systematic reviews. The absence of a standardised definition for nutrition therapy poses an additional challenge. Here, we based our definition on the American Diabetes Association,[13] but note that none of the eligible trials had interventions where nutrient or whole-food intake was directly manipulated, as seen in HICs[57] nor were the the interventions delivered by registered dietitians. Growing evidence points to the important role of community health workers for health promotion in LMICs[6 24] and to adapt research designs to the setting in which the interventions are to be implemented.[58] Being too restrictive in eligibility criteria when synthesising the evidence may hinder advances in evidence-guided improvements in care.

## CONCLUSIONS

Very low certainty of evidence impedes conclusions to be drawn on the impact of nutrition education on glycaemic control in people with T2DM living in LMICs. Even less is known about other types of nutrition therapy if we seek outcome data at least 3 months post-intervention. As T2DM becomes a growing problem in these countries, greater efforts are needed to build capacity for high-quality, context-appropriate and long-term research in lowest-income countries.

**Acknowledgements** We thank Prof. Simeon Pierre Choukem (University of Dschang, Dschang, Cameroon) for his insight on nutrition interventions in Cameroon for patients with type two diabetes mellitus, and for encouraging us to pursue this project.

**Contributors** SS and EG conceptualised the idea and led the review process. EG generated the first draft of the manuscript, and SS further refined it to adhere to the latest guidance on evaluation of the quality of evidence (ie, GRADE). RP led the design of the review methodology. AW guided the statistical aspects of the review and was responsible for the standardisation of the data. Both RP and AW edited the manuscript in accordance with their expertise. All authors approved the final version of the manuscript and agree to be accountable for all aspects of the work. SS is the guarantor and is overall responsible for this work.

**Funding** SS, RP and AW are supported by the NIHR Biomedical Research Centre at University Hospitals of Bristol and Weston NHS Foundation Trust and the University of Bristol.

**Disclaimer** The views expressed are those of the authors and not necessarily those of the NIHR or the Department of Health and Social Care.

**Competing interests** None declared.

**Patient consent for publication** Not applicable.

**Ethics approval** This study does not involve human participants.

**Provenance and peer review** Not commissioned; externally peer reviewed.

**Data availability statement** All data relevant to the study are included in the article or uploaded as supplementary information. Not applicable.

**ORCID iDs**
Rachel Perry http://orcid.org/0000-0001-5874-3016
Sarah Sauchelli http://orcid.org/0000-0003-3620-7671

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
