## [Reviewer comments · BMJ Open]

ARTICLE DETAILS

TITLE (PROVISIONAL)	Short-term effectiveness of nutrition therapy to treat type 2 diabetes in low- and middle-income countries: systematic review and meta-analysis of randomised-controlled trials.
AUTHORS	Guilbert, Elizabeth; Perry, Rachel; Whitmarsh, Alex; Sauchelli, Sarah

VERSION 1 – REVIEW

REVIEWER	Yamada, S Kitasato Daigaku, Kitasato Institute Hospital, Diabetes Center
REVIEW RETURNED	18-Sep-2021

GENERAL COMMENTS	This is a systematic review and meta-analysis of nutritional intervention trial in low- and middle-income countries (LMICs). According to their results, authors concluded that nutritional therapy alone may not be sufficient to improve glycemic control of T2DM patients in LMICs. But I believe nutritional intervention alone can improve glycemic control of T2DM patients even in LMICs. Thus, I feel authors' interpretation of their results seems too much pessimistic. I want authors to discuss with factors which concealed the effect of nutritional intervention in their study. For example, figures 3 and 4 suggest only 1 study by Ramadas et al. reported obviously negative result. In this study, within group analysis showed statistically significant improvement of HbA1c (from 9.1±2.0% to 8.7±1.9%) in intervention group. Surprisingly, there was huge improvement of HbA1c in control group (from 8.9±1.9% to 8.3±2.1%). I think this study showed the usefulness of treatment of control group, but not invalidity of nutritional intervention. Please discuss how to maximize the effect of nutritional intervention in LMICs. In addition, references 1-3 in page 4, line 2 seem not appropriate. As authors described the global diabetes situation here, only reference 1 is fit. References 1,2,4,5 in page 4, line3 are similar. As authors described the global diabetes burden here, only references 1 and 5 are fit. On the other hand, as authors described diabetes situation in LMICs, authors should refer references 2 and 4 in page 4, line 5.
---

REVIEWER	Nakajima, Kei Kanagawa University of Human Services, School of Nutrition and Dietetics, Faculty of Health and Social Services
-----------------	--

REVIEW RETURNED	26-Sep-2021
-------------

GENERAL COMMENTS	This study aims to evaluate the evidence arising from randomized controlled trials regarding the impact of nutritional therapy on glycemic control in people living with T2DM in LMICs. The authors concluded that in the randomized controlled trials with a 3-month follow-up assessment, nutritional therapy alone does not improve glycemic control in people living with T2DM in LMICs. This article may be of clinical interest. However, several issues should be improved before the consideration for publication. Major comments 1 The interventions for improving glycemic control are unclear. Although the authors described “Nutritional therapy” in the introduction, the description of nutrition education was observed in the methods and results section. If the nutrition education is a main method, the “Nutritional therapy” should be replaced with nutrition education throughout the article. In addition, the detail of nutrition education is also unclear. 2 I wonder that diabetes was poorly controlled in the 4 trials because the baseline HbA1c was very higher (10.8% and 11.4%). Such patients with very high HbA1c may be unsuitable for nutritional therapy. Pharmacotherapy may be considered as a first line therapy in most cases. 3 Many listed tables can be combined into one table, which should be shown in the text but not supplementary file. 4 Most of parameters in Tables are not accompanied with units. Please provide the corresponding unit. 5 The data of diet and physical activity are not provided and considered in the study. 6 The conclusion in the text should be concise. Minor comments 1 In the Abstract, is the description of “(MD): -0.63%, 95% CI -1.46 to 0.21) or fasting blood glucose (MD: -0.63%, 95% CI -1.46 to 0.21)” correct?
--

REVIEWER	Treadwell, Jonathan ECRI, Technology Assessment Group
REVIEW RETURNED	07-Oct-2021

GENERAL COMMENTS	BMJ review: Short-term effectiveness of nutritional therapy to treat type 2 diabetes in low- and middle-income countries: systematic review of randomized-controlled trials Jonathan R Treadwell, ECRI, Plymouth Meeting PA USA, October 2021 My comments are divided into major and minor to assist the BMJ Major comments The A1c meta-analytic confidence interval was as high as 1.46 for A1c and also 1.46 for FPG. Thus the evidence is consistent with
--

their being a benefit of nutrition therapy as high as 1.46 improvement in A1c. So it's surprising you conclude that there was no important effect. I would think, in order to have such a conclusion of approximate equivalence, your CI would have to be narrow enough to rule out the possibility of a minimally important difference in either direction, such as 0.4 on A1c. Your confidence is much wider than that, suggesting that a more appropriate statement would be "evidence is insufficient to determine whether there is an impact" rather than your conclusion that evidence is sufficient to conclude that there is no important impact.

In Supplementary Material 3, it appears that 3 of 4 four trials (all except Ramadas) used multicomponent interventions, and only one of the components was nutrition education. Thus it is hard to determine the impact of nutrition education alone. One of them even had people start jogging. Some had group sessions, and of course getting together with people like you could improve outcomes even with no nutrition intervention. What part of your risk of bias assessment considered the # of components in the intervention? This should be discussed. Authors stated that in each trial the "main component" was nutrition education, but didn't otherwise discuss the issue, as far as I could see. I think it warrants a few sentences in the methods.

Interesting Ramadas was the only study that had effect sizes that leaned against nutritional therapy. Perhaps this is related to the fact that that it was the only single-component-intervention trial, as far as I could tell from Supp Mat 3.

Authors said the intervention did not "meaningfully" impact outcomes. Did they define what level of impact they would have deemed meaningful? GRADE suggests reviewers set thresholds for small/moderate/large effects, and the certainty rating is one's "Certainty that a true effect lies on one side of a specified threshold, or within a chosen range". I understand it is hard to know where to draw these lines. For A1c at least, we've seen 0.3 or 0.4 as a typical threshold of importance. Probably there's one for FPG as well. You might create a new section in the methods called Data Interpretation which would include thresholds as well as GRADE methods.

Minor comments

Abstract says maximum treatment duration 8 weeks, and then says 3 month followup assessment. Since some readers might think that 3 months means after the START of the intervention, consider changing "follow-up" to "post-intervention".

Good idea to require 3 months+ post intervention, in order to be consistent with your interest in A1c.

Quite well-written in general

You excluded "metabolic syndrome without definitive T2DM diagnosis" and I'm wondering if there were any RCTs that were excluded for that reason alone. Terminology may not be used consistently in different countries and so there could be a relevant excluded RCT.

Under Data Analysis, in the section on meta-analysis, state whether you planned in advance to do random effects meta-analysis, or whether you might have done fixed effects analysis. Your forest plots are all random-effects.

	You say “First, we pooled studies according to the time at which the follow-up assessment was undertaken (3 vs 6 months).” Perhaps “3-6” months is less confusing since “vs” suggests either you compared 3 months to 6 months (which you didn’t) or you combined a given studies 3 month data with its 6 month data (which I doubt you did since you only report 1 timepoint per study). Also perhaps say “ we also performed separate analyses at both 3 months and 6 months”. Tau is a more direct measure of heterogeneity than I². See the article by Rucker in BMC Med Res Methodol 8(1) p79. The problem with I² is that it depends on the Ns. Try it yourself: take a meta-analysis and look at I². Now triple the Ns, but keep all the effect sizes the same. See how I² increased? Yeah. That’s not good. Note that tau would not have increased, since it more purely measures the extent to which effect sizes differed. I do understand that the Cochrane handbook seems to recommend I² and says “Methods have been developed for quantifying inconsistency across studies that move the focus away from testing whether heterogeneity is present to assessing its impact on the meta-analysis”. The handbook does however acknowledge tau (“the extent of variation, or heterogeneity, among the intervention effects observed in different studies (this variation is often referred to as Tau-squared, τ^2, or Tau2).”) I don’t have an easy answer for you other than to ask yourself, do you care more about measuring heterogeneity (tau) or do you care more about measuring the impact of heterogeneity on a meta-analysis (I²). I would think systematic reviewers more typically care about the former, and the former also is closer to what GRADE seems to like as an input to the inconsistency judgment. You also TESTED for homogeneity... doesn’t the Cochrane handbook recommend against testing? “Thus, the test for heterogeneity is irrelevant to the choice of analysis; heterogeneity will always exist whether or not we happen to be able to detect it using a statistical test.” It doesn’t actively HURT to test for it, but I’m wondering why bother. You did random effects models anyway so it’s not like you would have use the result of the chi square test to decide whether to present fixed or random.
--	--

VERSION 1 – AUTHOR RESPONSE

Reviewer: 1

Dr. S Yamada, Kitasato Daigaku

1. This is a systematic review and meta-analysis of nutritional intervention trial in low- and middle-income countries (LMICs). According to their results, authors concluded that nutritional therapy alone may not be sufficient to improve glycemic control of T2DM patients in LMICs. But I believe nutritional intervention alone can improve glycemic control of T2DM patients even in LMICs. Thus, I feel authors’ interpretation of their results seems too much pessimistic. I want authors to discuss with factors which concealed the effect of nutritional intervention in their study.

For example, figures 3 and 4 suggest only 1 study by Ramadas et al. reported obviously negative result. In this study, within group analysis showed statistically significant improvement of HbA1c (from 9.1±2.0% to 8.7±1.9%) in intervention group. Surprisingly, there was huge improvement of HbA1c in control group (from 8.9±1.9% to 8.3±2.1%). I think this study showed the usefulness of treatment of

control group, but not invalidity of nutritional intervention. Please discuss how to maximize the effect of nutritional intervention in LMICs.

Response: Upon revision of our manuscript, we recognise that the tone of the manuscript may have been too pessimistic. One key observation is that the four trials identified focused on nutrition education. According to the American Diabetes Association's definition of Nutrition Therapy (definition used in for this research project; Evert et al., 2013 <https://doi.org/10.2337/dc13-2042>), nutrition education is only one component of nutrition therapy. In agreement with the reviewer, meal replacement and other nutritional interventions play an important role in achieving glycaemic control. To avoid confusion we have adjusted the manuscript to emphasise why we may not have observed an effect in this systematic review, and in the discussion we provide possible explanations for the differential outcomes to previous work.

“The discrepancy, however, may be attributed to numerous factors. First, there is a lot of heterogeneity in the types of nutritional interventions tested in LMICs [5], possibly because they have been adapted for the target setting and resources available. As this review examined outcomes at 3 and 6 months after intervention completion (vs at completion) and all eligible trials focused on nutrition education (vs other types of nutrition therapy), comparability of findings is diminished. The emphasis on a 3 month post-intervention assessment is a strength of this review over others, as HbA1c provides an aggregate measure of the previous 3 months. Second, participants' baseline Hba1c levels in three trials were above the 10% threshold used to recommend insulin therapy initiation[46], or combining lifestyle interventions with pharmacotherapy [47], but there was no information regarding participants' use of pharmacotherapy. Third, the HbA1c levels of the control group in one study improved quite notably, which could lessen the relative effectiveness of the intervention and influenced the effect estimate. Nonetheless, the wide confidence intervals in the effect estimate indicate that the true effect may be clinically meaningful and accrual of research could provide increased confidence in estimates.” (page 16).

Further, we have provided other components of nutritional therapy as defined by the American Diabetes Association that could help maximise the effect of nutritional interventions in LMICs, adding to our recommendation for a multi-faceted approach to diabetes management strategies. We have added the following statement

“A key consideration for clinicians and policy makers is that the American Diabetes Association's definition of nutrition therapy, encompasses nutrition education and interventions that yield modification of nutrient or whole-food intake [13]. Tailoring provision of nutrition guidance to the individual, delivery by an expert in diabetes care, and ongoing monitoring to permit modification as needed have been identified as fundamental features of effective nutrition therapy in type 2 diabetes care [13]. These components of nutrition therapy were not consistently present in identified trials but could strengthen observed effectiveness of nutrition therapy.” (page 16)

Finally, throughout the discussion we have adapted our message to emphasise the need for more research and consideration of real-world research when deriving conclusions and informing clinical practice.

“Very low certainty of evidence impedes conclusions to be drawn on the impact of nutrition education on glycaemic control in people with T2DM living in LMICs. Even less is known about other types of nutrition therapy if we seek outcome data at least 3 months post-intervention. As T2DM becomes a growing problem in these countries, greater efforts are needed to build capacity for high-quality, context-appropriate, and long-term research in lowest-income countries.” (page 18)

2. In addition, references 1-3 in page 4, line 2 seem not appropriate. As authors described the global diabetes situation here, only reference 1 is fit.

Response: We have removed references 2 and 3 as guided by the reviewer.

3. References 1,2,4,5 in page 4, line3 are similar. As authors described the global diabetes burden here, only references 1 and 5 are fit.

Response: We have removed references 2 and 4 as recommended by the reviewer.

4. On the other hand, as authors described diabetes situation in LMICs, authors should refer references 2 and 4 in page 4, line 5.

Response: We have added references 2 and 4 here as recommended by the reviewer.

Reviewer: 2

Prof. Kei Nakajima, Kanagawa University of Human Services

Comments to the Author:

This study aims to evaluate the evidence arising from randomized controlled trials regarding the impact of nutritional therapy on glycemic control in people living with T2DM in LMICs.

The authors concluded that in the randomized controlled trials with a 3-month follow-up assessment, nutritional therapy alone does not improve glycemic control in people living with T2DM in LMICs.

This article may be of clinical interest.

However, several issues should be improved before the consideration for publication.

Major comments

1. The interventions for improving glycemic control are unclear. Although the authors described “Nutritional therapy” in the introduction, the description of nutrition education was observed in the methods and results section. If the nutrition education is a main method, the “Nutritional therapy” should be replaced with nutrition education throughout the article. In addition, the detail of nutrition education is also unclear.

Response: The author is correct in that in the manuscript there is a shift towards nutrition education. The reason for this is that our screening of RCTs with a post-intervention assessment of glycaemic control 3 months after yielded only 4 eligible trials, and their delivered nutrition education, which is only one component of nutrition therapy. We have maintained the title and introduction as “nutrition therapy” because this is what we sought out to do, and our search strategy was developed accordingly. However, we have made the following changes to the manuscript:

a) Abstract, Results and Discussion – emphasised that all trials considered as eligible focussed on nutrition education, and none looked at the impact of direct manipulation or prescription of diet. e.g. “All trials focussed on nutrition education with no direct prescription or manipulation of diet.” (page 2)

b) We have better defined nutrition therapy in the Introduction:

“Nutrition therapy was defined according to the consensus recommendations presented by the American Diabetes Association for diabetes and pre-diabetes [13], where treatment entails the modification of nutrient or whole-food intake. This definition included interventions that provide individualised, diabetes-focussed management plan and ongoing monitoring, that address individual nutrition needs and that distribute positive messaging about food choices, as well as nutrition education, lifestyle intervention programmes with goal-setting, provision of tools for day-to-day meal planning.” (page 4)

c) We have provided a more in-depth description of how each study evaluated nutrition education: “All RCTs focussed on nutrition education to prompt people to make healthy dietary choices and improve portion control. There was no direct manipulation of nutrient or whole-food intake (e.g. prescribing a specific diet, meal replacement, or providing food supplements/substitutes). In 1 trial participants received a pamphlet and fridge/wall poster in line with standard care, complemented by an 8-week face-to-face group (6-8 people) nutrition education programme (dietitian-led) that targeted knowledge deficits and inappropriate dietary practices, and 6 follow-up sessions [39]. The 8 sessions covered the following topics: definition of diabetes, treatment, dietary guidelines (2 sessions), gardening to increase vegetable supply, meal planning (2 sessions) and meal preparation [39]. 1 trial provided dietary lesson plans (12 sessions, fortnightly) delivered via a website and tailored to each participant’s current Dietary Stage of Change (DSOC, a system to classify an individual’s readiness to change a health-related behaviour)[40]. The sessions included tailored recommendations aimed at improving diabetes knowledge and behaviour, address barriers to dietary change, and motivate participants. 1 trial delivered educational sessions delivered by a range of health specialists (e.g. dietitians, psychologists) in the meeting hall of a healthcare centre. These were aimed at addressing perceptions, beliefs, fears, concerns and discomforts towards a diabetic diet (e.g., based on psychological factors which determine nutrition such as avoiding temptation, communication skills training, replacing irrational thinking, religious laws around health and focusing on success) in the form of 8 one-hour sessions with groups of 23-24 participants [41]. The fourth trial delivered a group educational intervention over 4 weeks (2 70 minute training sessions per week) based on the Belief, Attitudes, Subjective Norm and Enabling Factors model; giving participants in the intervention arm information about diabetes, symptoms of hyper- and hypo-glycaemia, diet, use of food composition tables, partitioning and food replacement and appropriate intake of fruits, vegetables, and grains (Askari et al., 2018). This trial was the only one to ask participants to record daily food consumption (though the use of those records could not be identified), conduct follow-up calls (2 calls), send weekly reminder text messages, and to include exercise in the form of a prescribed jogging regime [42].” (page 8)

d) Our conclusions distinguishes between what we know regarding nutrition education, and the lack of information regarding other types of nutrition therapy.

“Very low certainty of evidence impedes conclusions to be drawn on the impact of nutrition education on glycaemic control in people with T2DM living in LMICs. Even less is known about other types of nutrition therapy if we seek outcome data at least 3 months post-intervention.” (page 18)

2. I wonder that diabetes was poorly controlled in the 4 trials because the baseline HbA1c was very higher (10.8% and 11.4%). Such patients with very high HbA1c may be unsuitable for nutritional therapy. Pharmacotherapy may be considered as a first line therapy in most cases.

Response: In agreement with the reviewer, various health organisation recommend pharmacotherapy parallel to nutrition therapy when glycaemic index is elevated, to enhance the effectiveness. We were unable to obtain information regarding drug treatments that participants may have been undertaking (if any), and therefore have added the following extract in the discussion:

“ Second, participants’ baseline Hba1c levels in three trials were above the 10% threshold used to recommend insulin therapy initiation[46], or combining lifestyle interventions with pharmacotherapy [47], but there was no information regarding participants’ use of pharmacotherapy.” (page 16)

3. Many listed tables can be combined into one table, which should be shown in the text but not supplementary file.

Response: As recommended by the reviewer we brought together the figures showing changes in HbA1c and fasting blood glucose (now Figure 3). We have additionally moved the table reporting characteristics of included studies and that reporting secondary outcomes from supplementary

materials to the main text (Tables 2 and 3 respectively). However, we are reluctant to move additional texts and figures given that BMJ Open strives to minimise number of tables/figures in papers.

4. Most of parameters in Tables are not accompanied with units. Please provide the corresponding unit.

Response: As requested by the reviewer we have added the units with which parameters were assessed across all tables. An example is Table 1 (pages 10-11).

5. The data of diet and physical activity are not provided and considered in the study.

Response: Thank you for highlighting that we did not include data on physical activity and diet. When designing and registering this review we were particularly interested in patient-important outcomes and their surrogates. We consulted both the existing literature and clinical experts to determine that glycaemic control should be a primary outcome as it is a key indicator of risk of diabetes-related complications (e.g., major morbid events, impaired daily functioning, visual disturbances, and pain). In addition to other indicators of possible negative life events (e.g., BMI, blood pressure, cholesterol and triglycerides) we opted to include the psychological effects that the intervention may have on mental health and quality of life. Hence, we included outcomes on perceived adherence to the diet and diabetes knowledge, but not diet itself. When considering whether to amend the registered protocol to add information about diet and physical activity we concluded that it would not yield additionally valuable information. Any clinically meaningful changes to energy intake and nutrient composition of daily diet, should appear in the glycaemic control and other surrogate outcomes (e.g. BMI). Further, though the Askari paper included two checklists to evaluate whether participants were adhering to nutrition and jogging recommendations, there was very little information about the content of those 6-item checklists (and scoring) making it difficult to interpret results on a face value. We made several attempts at contacting the authors for further clarifications but did not receive a reply.

Nonetheless, we acknowledge the comment made by reviewer as valid and have included a statement in the discussion regarding the value of measuring and reporting this information in primary studies, and summarising it wherever possible in systematic reviews on the topic.

“Another weakness of this review is that we were unable to summarise data on actual diet and physical activity. Although neither were in the pre-registered protocol as we focused on surrogate measures of patient important outcomes (e.g. HbA1c, fasting blood glucose, and BMI), dietary behaviour and physical activity should be measured and reported by RCTs on this subject, and the information should be measured in RCTs and reported in future systematic reviews.” (page 17-18)

5. The conclusion in the text should be concise.

By making changes to the manuscript guided by reviewers' comments, the conclusions is now more concise:

“Very low certainty of evidence impedes conclusions to be drawn on the impact of nutrition education on glycaemic control in people with T2DM living in LMICs. Even less is known about other types of nutrition therapy if we seek outcome data at least 3 months post-intervention. As T2DM becomes a growing problem in these countries, greater efforts are needed to build capacity for high-quality, context-appropriate, and long-term research in lowest-income countries.” (page 18)

Minor comments

1 In the Abstract, is the description of “(MD): -0.63%, 95% CI -1.46 to 0.21) or fasting blood glucose (MD: -0.63%, 95% CI -1.46 to 0.21)” correct?

Response: We thank the reviewer for noticing this important error. The abstract has been changed to the following: "(MD): -0.63%, 95% CI -1.47 to 0.21) or fasting blood glucose (MD: -13.63 mg/dl, 95% CI -37.61 to 10.34)" (page 2)

Reviewer: 3

Dr. Jonathan Treadwell, ECRI

Comments to the Author:

BMJ review: Short-term effectiveness of nutritional therapy to treat type 2 diabetes in low- and middle-income countries: systematic review of randomized-controlled trials

Jonathan R Treadwell, ECRI, Plymouth Meeting PA USA, October 2021

My comments are divided into major and minor to assist the BMJ

Major comments

The A1c meta-analytic confidence interval was as high as 1.46 for A1c and also 1.46 for FPG. Thus the evidence is consistent with their being a benefit of nutrition therapy as high as 1.46 improvement in A1c. So it's surprising you conclude that that there was no important effect. I would think, in order to have such a conclusion of approximate equivalence, your CI would have to be narrow enough to rule out the possibility of a minimally important difference in either direction, such as 0.4 on A1c. Your confidence is much wider than that, suggesting that a more appropriate statement would be "evidence is insufficient to determine whether there is an impact" rather than your conclusion that evidence is sufficient to conclude that there is no important impact.

Response: We thank the reviewer for this valuable comment. Upon further review of the evidence and consultation with our clinical colleagues, it seems that a decrease by 0.5% or 1% in HbA1c concentrations can be considered as a clinically meaningful reduction following treatment (e.g. Leters-Westra et al. 2014; Netherlands Journal of Medicine, 72:462-466; Selvin et al. 2004; Annals of Internal Medicine, 141:6). Hence, if the true value were to lie anywhere between -1% and 0.5%, we would conclude that nutrition education generates a clinically important change. Although the threshold for concluding a clinically meaningful change in fasting blood glucose (FBG) is less clear, there is evidence that a reduction of 18 mg/dL can reduce risk of vascular disease (Sarwar et al. 2010 Lancet, 375,2215). Again, this value falls within the CIs. Hence, in response to the reviewer to this and a latter comment we have made the following changes:

a) Introduced a 'Data Interpretation' subsection in the methods.

"There are no clear-cut thresholds to conclude that an intervention under evaluation is superior to standard care on the basis of HbA1c or fasting blood glucose. Previous work suggests that a reduction by 0.5% or 1% in HbA1c is often used by health professionals when making adjustments to therapy [35] and is beneficial for reducing cardiovascular disease risk, a patient important outcome [36]. Given that fasting blood glucose is often a secondary end point in RCTs, establishing a threshold for meaningful effect was also derived from vascular risk, which suggests a threshold value of one 18 mg/l[37]." (page 7)

b) Included a statement in the results.

"The point estimates for both HbA1c and fasting blood glucose include both the line of "no effect" and the threshold for concluding clinically meaningful difference."(page 12)

c) Included a statement in the discussion.

“Nonetheless, the wide confidence intervals in the effect estimate indicate that the true effect may be clinically meaningful and accrual of research could provide increased confidence in estimates.” (page 16)

d) Adapted the text to reflect that there may be a clinically important difference, there is uncertainty in the available evidence, and more evidence needs to be collected to increase certainty on the true effect.

e.g. “ The advantage was less clear in trials with a 6-month post-intervention assessment, with pooled mean differences at -0.09% (95% CI -1.10 to 0.91) for HbA1c and 9mg/dl (95% CI -7.55 to 25.55) for fasting blood glucose. Similar uncertainty is present when examining all trials jointly, with pooled mean differences at -0.63% (95% CI -1.46 to 0.21; 4 trials, n=463, GRADE=very low) for HbA1c and -13.63mg/dl (95% CI -37.61 to 10.34; 3 trials, n=381, GRADE=very low) for fasting blood glucose.” (page 12)

In Supplementary Material 3, it appears that 3 of 4 four trials (all except Ramadas) used multicomponent interventions, and only one of the components was nutrition education. Thus it is hard to determine the impact of nutrition education alone. One of them even had people start jogging. Some had group sessions, and of course getting together with people like you could improve outcomes even with no nutrition intervention. What part of your risk of bias assessment considered the # of components in the intervention? This should be discussed. Authors stated that in each trial the “main component” was nutrition education, but didn’t otherwise discuss the issue, as far as I could see. I think it warrants a few sentences in the methods.

Interesting Ramadas was the only study that had effect sizes that leaned against nutritional therapy. Perhaps this is related to the fact that that it was the only single-component-intervention trial, as far as I could tell from Supp Mat 3.

Response: We have provided further details of the interventions of included trials in the Results section (page 8) and clarified intervention summaries presented in the “Characteristics of included trials” table (now Table 1). As we hope these changes will clarify, there was a lot heterogeneity in the interventions, with 2 providing only nutrition education (Ramadas and Salashouri), 2 adding follow-up sessions and/or text messages to reinforce the effectiveness of training (Muchiri, Askari), and 1 encouraging more physical activity (Askari). Unfortunately, due to the limited number of trials identified as eligible and unclear reporting we were unable to further explore whether multi-component interventions provide superior result, despite much of the literature and non-RCTs suggesting this might be the case. Hence, we have added a statement in the discussion on this limitation.

“A weakness of this review is that the small number of trials identified did not permit sub-group analysis by separating interventions that consisted only in nutrition education from the ones with additional components (e.g. follow-up sessions, exercise).” (page 17)

After careful review of the Cochrane guidance on risk of bias and the CRIBSHEET provided by Prof. Julian Higgins and his team, we also believe that given the nature of the components in the included RCTs and that the outcome measures were derived from blood specimen, risk of bias judgements would not be impacted in this case. However, we emphasise in the paper the importance of taking into account the various components within an intervention when assessing risk of bias:

“Further, risk of bias for each outcome was assessed by considering the intervention delivered in each trial in its entirety. Although we would not expect differences in individual study risk of bias assessments based on the additional components described, future research should consider the

possible impact of evaluating a multicomponent intervention (e.g. adding nutrition education to pharmacotherapy) on risk of bias.” (page 17)

We have also provided further information on what we mean by “main component” in the methods section:

“Trials were included when the main component was nutrition therapy (i.e., diet modification) achieved by directly prescribing a diet, meal replacement or use of supplements or by encouraging change via nutrition education. Where nutrition therapy was not the most influential component expected to contribute to changes in glycaemic control (e.g. one structured nutrition education session within a two year exercise programme, a short course of meal replacement combined with long-term metformin prescription), the RCT was excluded.” (page 6)

Authors said the intervention did not “meaningfully” impact outcomes. Did they define what level of impact they would have deemed meaningful? GRADE suggests reviewers set thresholds for small/moderate/large effects, and the certainty rating is one’s “Certainty that a true effect lies on one side of a specified threshold, or within a chosen range”. I understand it is hard to know where to draw these lines. For A1c at least, we’ve seen 0.3 or 0.4 as a typical threshold of importance. Probably there’s one for FPG as well. You might create a new section in the methods called Data Interpretation which would include thresholds as well as GRADE methods.

Response: We thank the reviewer for this suggestion, and as reported in response to an earlier comment, we have added a section in the Methods titled “Data Interpretation”, where we provide our thresholds for clinically meaningful difference. We have also adjusted results reporting and discussion to provide a more conservative interpretation of data synthesis.

Minor comments

1. Abstract says maximum treatment duration 8 weeks, and then says 3 month followup assessment. Since some readers might think that 3 months means after the START of the intervention, consider changing “follow-up” to “post-intervention”.

Response: We thank the reviewer for this recommendation and have changed the abstract accordingly (page 2).

2. Good idea to require 3 months+ post intervention, in order to be consistent with your interest in A1c.

Response: Thank you

3. Quite well-written in general

Response: Thank you

4. You excluded “metabolic syndrome without definitive T2DM diagnosis” and I’m wondering if there were any RCTs that were excluded for that reason alone. Terminology may not be used consistently in different countries and so there could be a relevant excluded RCT.

Response: The reviewer makes a valid point. Despite the improvements in how T2DM is defined, diagnosed and reported across the world, there are studies referring to metabolic syndrome, when either the whole study or over 40% of the sample had T2DM. We reviewed all abstracts excluded where the reason provided was ‘wrong population’ and we did not detect any that were excluded solely for using the term ‘metabolic syndrome’. There were a few cases where the participant sample comprised people with ‘Metabolic Syndrome’ but were excluded on this basis because they evaluated people with a mixture of sub-diagnoses (e.g. T2DM, obesity etc.) and the sample with T2DM comprised less than 20% of total sample.

5. Under Data Analysis, in the section on meta-analysis, state whether you planned in advance to do random effects meta-analysis, or whether you might have done fixed effects analysis. Your forest plots are all random-effects.

Response: We planned in advance to conduct random-effects meta-analyses as our expectations on the types of RCTs we would identify suggested that we would need to relax the assumption of statistical homogeneity. We have added the following statement in the methods:

“As we expected both clinical and methodological variation between eligible studies of which cause would be difficult to identify, random effect analyses were considered as most appropriate.” (page 7)

6. You say “First, we pooled studies according to the time at which the follow-up assessment was undertaken (3 vs 6 months).” Perhaps “3-6” months is less confusing since “vs” suggests either you compared 3 months to 6 months (which you didn’t) or you combined a given studies 3 month data with its 6 month data (which I doubt you did since you only report 1 timepoint per study). Also perhaps say “ we also performed separate analyses at both 3 months and 6 months”.

Response: Thank you for the suggestion. We have made the changes to the methods section as recommended by the reviewer.

“First, we pooled trials according to the time at which the post-intervention assessment was undertaken (3-6 months), and performed separate analyses at both 3 months and 6 months.” (page 7)

7. Tau is a more direct measure of heterogeneity than I². See the article by Rucker in BMC Med Res Methodol 8(1) p79. The problem with I² is that it depends on the Ns. Try it yourself: take a meta-analysis and look at I². Now triple the Ns, but keep all the effect sizes the same. See how I² increased? Yeah. That’s not good. Note that tau would not have increased, since it more purely measures the extent to which effect sizes differed. I do understand that the Cochrane handbook seems to recommend I² and says “Methods have been developed for quantifying inconsistency across studies that move the focus away from testing whether heterogeneity is present to assessing its impact on the meta-analysis”. The handbook does however acknowledge tau (“the extent of variation, or heterogeneity, among the intervention effects observed in different studies (this variation is often referred to as Tau-squared, τ^2 , or Tau2).”) I don’t have an easy answer for you other than to ask yourself, do you care more about measuring heterogeneity (tau) or do you care more about measuring the impact of heterogeneity on a meta-analysis (I²). I would think systematic reviewers more typically care about the former, and the former also is closer to what GRADE seems to like as an input to the inconsistency judgment.

Response: We thank the reviewer for the clarification. After carefully reading the appropriate sections in the Cochrane handbook and GRADE guidelines we have opted for reporting both (visible in Figure 3).

8. You also TESTED for homogeneity...doesn’t the Cochrane handbook recommend against testing? “Thus, the test for heterogeneity is irrelevant to the choice of analysis; heterogeneity will always exist whether or not we happen to be able to detect it using a statistical test.” It doesn’t actively HURT to test for it, but I’m wondering why bother. You did random effects models anyway so it’s not like you would have use the result of the chi square test to decide whether to present fixed or random.

Response: We opted for reporting X² to be consistent with other studies. However, we recognise that interpretation of the P-value to evaluate indirectness (as suggested by Guyatt et al., 2011; Journal of Clinical Epidemiology; 64(12):1294-1302) when the number of studies is small can be misleading. Our judgement on the indirectness domain was therefore guided by multiple factors.

Reviewer: 1
Competing interests of Reviewer: Nothing to declare

Reviewer: 2
Competing interests of Reviewer: None

Reviewer: 3
Competing interests of Reviewer: None

VERSION 2 – REVIEW

REVIEWER	Yamada, S Kitasato Daigaku, Kitasato Institute Hospital, Diabetes Center
REVIEW RETURNED	25-Jan-2022
GENERAL COMMENTS	Authors have responded to reviewers' comments well in their revised manuscript.